# Understanding the relationship between cerebellum and the frontal-cortex region of C9orf72-related amyotrophic lateral sclerosis: A comparative analysis of genetic features

**Kartikay Prasad**[1], **Md Imtaiyaz Hassan**[2], **Saurabh Raghuvanshi**[3]*, **Vijay Kumar**[1]*

**1** Amity Institute of Neuropsychology & Neurosciences, Amity University, Noida, UP, India, **2** Centre for Interdisciplinary Research in Basic Sciences, Jamia Millia Islamia, New Delhi, India, **3** Department of Plant Molecular Biology, University of Delhi, South Campus, New Delhi, India

* vkumar33@amity.edu (VK); saurabh@genomeindia.org (SR)

**Data Availability Statement:** All relevant data are within the manuscript and its Supporting Information files.

## Abstract

### Background

Amyotrophic lateral sclerosis (ALS) is a relentlessly progressive and fatal neurodegenerative diseases for which at present no cure is available. Despite the extensive research the progress from diagnosis to prognosis in ALS and frontotemporal dementia (FTD) has been slow which represents suboptimal understanding of disease pathophysiological processes. In recent studies, several genes have been associated with the ALS and FTD diseases such as SOD1, TDP43, and TBK1, whereas the hexanucleotide GGGGCC repeat expansion (HRE) in *C9orf72* gene is a most frequent cause of ALS and FTD, that has changed the understanding of these diseases.

### Methods

The goal of this study was to identify and spatially determine differential gene expression signature differences between cerebellum and frontal cortex in C9orf72-associated ALS (C9-ALS), to study the network properties of these differentially expressed genes, and to identify miRNAs targeting the common differentially expressed genes in both the tissues. This study thus highlights underlying differential cell susceptibilities to the disease mechanisms in C9-ALS and suggesting therapeutic target selection in C9-ALS.

### Results

In this manuscript, we have identified that the genes involved in neuron development, protein localization and transcription are mostly enriched in cerebellum of C9-ALS patients, while the UPR-related genes are enriched in the frontal cortex. Of note, UPR pathway genes were mostly dysregulated both in the C9-ALS cerebellum and frontal cortex. Overall, the data presented here show that defects in normal RNA processing and the UPR pathway are the pathological hallmarks of C9-ALS. Interestingly, the cerebellum showed more strong transcriptome changes than the frontal cortex.

**Funding:** K.P. sincerely thanks the Indian Council of Medical Research (ICMR) New Delhi, India for providing the Senior Research Fellowship grant (BMI/11(63)/2020). VK sincerely thanks Indian Council of Medical Research (ICMR) New Delhi, India (Grant No. DHR/Neuro/2020-NCD-I). The funders had no role in study design, data collection and analysis, decision to publish, or preparation of the manuscript.

## Conclusion

Interestingly, the cerebellum region showed more significant transcriptomic changes as compared to the frontal cortex region suggesting its active participation in the disease process. This nuanced understanding may offer valuable insights for the development of targeted therapeutic strategies aimed at mitigating disease progression in C9-ALS.

## Introduction

Amyotrophic lateral sclerosis (ALS) is a chronic age-related neurodegenerative disorder that affects upper motor neurons (UMN) and lower motor neurons (LMN) in the motor cortex, brainstem, and spinal cord, eventually leading to paralysis and death, usually due to respiratory failure, within 2–5 years of symptom onset [1,2]. ALS is a disease with multiple risk factors, including genetic, epigenetic, and environmental variables [3]. ALS has been linked to more than 50 possibly causative or disease-modifying genes; however, pathogenic mutations in SOD1, C9orf72, FUS, and TARDBP are the most common [4–8]. Of all the ALS cases reported approximately 90% of the cases belongs to sALS (sporadic ALS) and the remaining cases belongs to fALS (familial ALS) [9]. The major pathogenic impact mutation can be detected in 60–80% of fALS patients, with C9orf72 (40%), SOD1 (20%), FUS (1–5%), and TARBDP (1–5%) being the most prevalent [10–12]. Several genetic and pathophysiological pathways contribute to the disease, and understanding this heterogeneity will be required to develop successful treatments [8,13]. The most common mutation associated with ALS and frontotemporal dementia (FTD) is the GGGGCC hexanucleotide repeat expansion (HRE) in the non-coding region of the C9orf72 gene [14,15]. The mutation is responsible for ~3%– 6% of sALS, 6% of sFTD, 10% –25% of fFTD, 20%– 40% of fALS [14,16,17] The most three pathomechanisms associated with HRE are: C9orf72 protein function loss, toxic gain-of-function due to RNA foci buildup, and synthesis of dipeptide repeat proteins via repeat-associated non-ATG (RAN) translation [14,18–22].

However, the clinical manifestations of the C9orf72 HRE mutation are highly variable, with people bearing the same mutation exhibiting a wide range of symptoms across the ALS-FTD spectrum [23]. This high clinical heterogeneity presents a significant diagnostic difficulty and makes ALS patient classification in clinical studies extremely difficult. Despite this clinical variation, common pathological pathways in the etiology of ALS exist [24].

The selection of frontal and cerebellar regions for ALS studies stems from their pivotal roles in the pathophysiology of the disease. The frontal cortex, implicated in cognitive functions and motor control, undergoes significant degeneration in ALS, contributing to cognitive impairment and motor dysfunction observed in patients [25]. Additionally, the cerebellum, traditionally associated with motor coordination, has emerged as a site of neurodegeneration in ALS, with studies implicating its involvement in disease progression and symptomatology [26]. Recent evidence suggests the cerebellum's involvement in C9orf72-associated ALS/FTD, with decreased C9ORF72 levels detected postmortem, indicating a potential loss-of-function effect [27,28]. Using a C9orf72 knockout mouse model, this study reveals the profound impact of C9ORF72 deficiency on motor function, attributed to hyperactive Purkinje cells and upregulated BK protein. Chemogenetic manipulation of Purkinje cells in wild-type mice mimics motor deficits seen in C9orf72 knockout mice, highlighting the cerebellum's pivotal role in disease pathogenesis [29].

By focusing on these regions, researchers can elucidate the molecular mechanisms underlying ALS pathology and identify potential therapeutic targets. Furthermore, the comparative

analysis of gene expression profiles between these regions and control samples can provide insights into region-specific alterations and their contribution to disease manifestation. This approach enhances our understanding of ALS pathology and facilitates the development of targeted interventions aimed at mitigating disease progression.

To better understand the mechanisms associated with clinical heterogeneity, different transcriptomic studies have been performed to understand the mechanisms of neurodegeneration [30–33]. Transcriptome investigation of ALS patients' frontal cortical and motor neurons revealed splicing alterations, dysregulation of RNA processing pathways, and alternative polyadenylation in sporadic and C9orf72 HRE carriers [30,34,35]. A large transcriptome study of C9orf72 cases also discovered changes in genes associated with vesicular transport [36]. These studies are sometimes complicated by the diverse biological composition of brain tissue, which includes neuronal subtypes, oligodendrocytes, microglia, astrocytes, vascular cells, and other cell types.

The goal of this study was to identify and spatially determine differential gene expression signature differences between cerebellum and frontal cortex in C9-ALS, to study the network properties of these differentially expressed genes, and to identify miRNAs targeting the common differentially expressed genes in both the tissues. This study thus highlights underlying differential cell susceptibilities to the disease mechanisms in C9-ALS and suggesting therapeutic target selection in C9-ALS.

## Material and methods

### Samples

Samples with RNA-sequence data were obtained from two different sources: Firstly, from the Target ALS Human Post-mortem Tissue Core and the New York Genome Centre (NYGC) published by Conlon et al. [31] (NCBI GEO ID: GSE116622) and Tam et al. [32] (GSE137810), and secondly from the Florida Mayo Clinic published by Prudencio et al. [30] (GSE67196). For RNA-Seq study of GSE137810, frozen human post-mortem tissue was obtained from frontal cortex and cerebellum regions. For GSE67196, frozen human post-mortem tissue was taken from the lateral hemisphere of the cerebellum, and the prefrontal cortex (Brodmann area 9/44) regions. And, for GSE116622, the samples were obtained from the New York Brain Bank. A total of 25 C9-ALS patients and 25 controls each from cerebellum and frontal cortex regions were downloaded from the NCBI Gene Expression Omnibus (GEO) database with GEO ids GSE67196, GSE116622, and GSE137810. ALS samples and controls were matched by age and sex where possible for each individual dataset.

### Data collection and quality check

SRA-toolkit was used to retrieve the data form the NCBI database [37]. Once the sequencing read was obtained, the first step was to check the quality of the reads using FastQC software [38]. FastQC is a quality control application for high throughput sequence data. It uses the quality score or phred score of the nucleotides to generate quality check files. Based on the quality checks result the trimming and filtering of the reads was done for the required samples. Removing artefacts from RNA-Seq data sets before assembly improves the read quality, which, in turn, improves the accuracy and computational efficiency of the assembly.

### Sequence mapping and generation of count data

After trimming and filtering process, the reads were aligned to the reference genome (Humans:GRch38) using the RSEM tool [39]. RSEM tool uses the bowtie2 software in the

background for genome referencing and reads mapping. The tool provides the output in bam format. Apart from the mapped reads, RSEM generates the count data matrix representing the number of reads mapped on each gene along with TPM (transcript per million) and FPKM (Fragments per kilobase of transcripts per million mapped reads) values.

## Differential gene expression analysis

After obtaining the count data of the control/healthy and the C9-ALS patient samples, the count data was then used to identify the differentially expressed genes (DEGs) across the cerebellum and frontal cortex samples. Before identifying the DEGs, one of the most crucial step is to perform the data normalization. Data normalization removes the technical biasedness inherent in the sequencing approach, most notably the length of RNA species and the sequencing depth of the sample. Differential gene expression of C9-ALS patient samples was identified in comparison with the normal patient samples using Dseq2 package of R language [40]. Dseq2 tests differential expression using negative binomial generalized linear models. Genes with fold-change $\geq$ 2, P-value $<$ = 0.01 and FDR $<$ = 0.01 were considered significantly dysregulated and were used further in the study. Based on log2transformed count values, unsupervised clustering of the dysregulated genes was performed using the "pheatmap" package of R. Further, genes commonly showing differential expression in both the tissues, i.e., cerebellum and frontal cortex, were identified using InteractiVenn, a web-based tool to analyse the gene sets through Venn diagram [41].

## Gene-set and pathway enrichment analysis

GSEA (Gene Set Enrichment Analysis) was performed for the comprehensive analysis of the identified differentially expressed genes. GSEA identifies the role of differentially expressed genes at biological, cellular and at molecular levels using the clusterProfiler programme of R language [42]. Further a pathway enrichment analysis was performed to identify the role of the differentially expressed genes in different pathways using KEGG database [43–45].

## Preparing brain specific gene-gene and disease-gene interaction network

The genes with elevated expression in the brain were retrieved from the Protein atlas database [46]. A gene-gene interaction network was prepared with the list of elevated genes and commonly identified dysregulated genes as input in the string plugin of the cytoscape tool [47,48]. Next, a subnetwork was prepared to identify the genes directly interacting with the commonly dysregulated genes in both the tissues. Further, a brain related disease-gene interaction network of the commonly dysregulated genes and their neighbouring genes was prepared by screening multiple Disease-gene interaction databases such as GeneORGANizer, DisGeNET and MalaCards to identify their roles in other brain related disorders [49–51].

## Identification of miRNA's as regulator

Several miRNA-gene interaction databases, including miRTarBase, miRbase, miRDB, and miRNet2, were analyzed to discover miRNAs interacting with commonly observed dysregulated genes in both tissues [52–55]. These databases also offered interaction validation information utilizing literature-based analysis. For further investigation, miRNA-gene interactions with at least one validation method were considered for further analysis. For the gene ontology and pathway-based enrichment analysis of the chosen miRNAs, the GeneTrail database was employed [56].

## Results and discussion

Recent advances in C9-ALS molecular genetics have revealed a link between RBP aggregation, defects in protein clearance pathway, and RNA processing dysfunction [57]. Despite the increased availability of transcriptomic data [30,31,57,58], an unresolved challenge remains in identifying the most central and meaningful gene expression changes that may drive the disease phenotype. The present investigation examined transcriptomic alterations in both the cerebellum and frontal cortex of individuals with C9-ALS, revealing a substantial convergence of dysregulated genes. This convergence implies the existence of shared pathological pathways in C9-ALS, underscoring the presence of common underlying mechanisms contributing to the disease across these distinct brain regions.

### Identification of differentially expressed genes

For identifying the differentially expressed genes in cerebellum and the frontal cortex region of the C9-ALS patient samples, first we mapped the sequencing reads on the reference genome using the RSEM tool and prepared a gene count matrix. Genes having -2< = log2foldChange < = 2, along with P-value and FDR < = 0.01 obtained from DESeq2 analysis were considered significantly dysregulated. A total of 117 and 127 genes were significantly dysregulated in the cerebellum and the frontal cortex samples, respectively [Figs 1A and 2A]. Out of the identified significantly dysregulated genes, 57 genes in cerebellum [Fig 1B] [S1 Table in S1 Data] and 71 genes in frontal cortex has the protein-coding properties [Fig 2B] [S2 Table in S1 Data], and the remaining genes belongs to the non-protein-coding categories such as long and small noncoding RNAs, and more.

### Gene-set and pathway enrichment analysis of differentially expressed genes

The clusterProfiler programme of R was used for the GSEA (Gene Set Enrichment Analysis) analysis of the identified differentially expressed genes from both the tissue samples. The

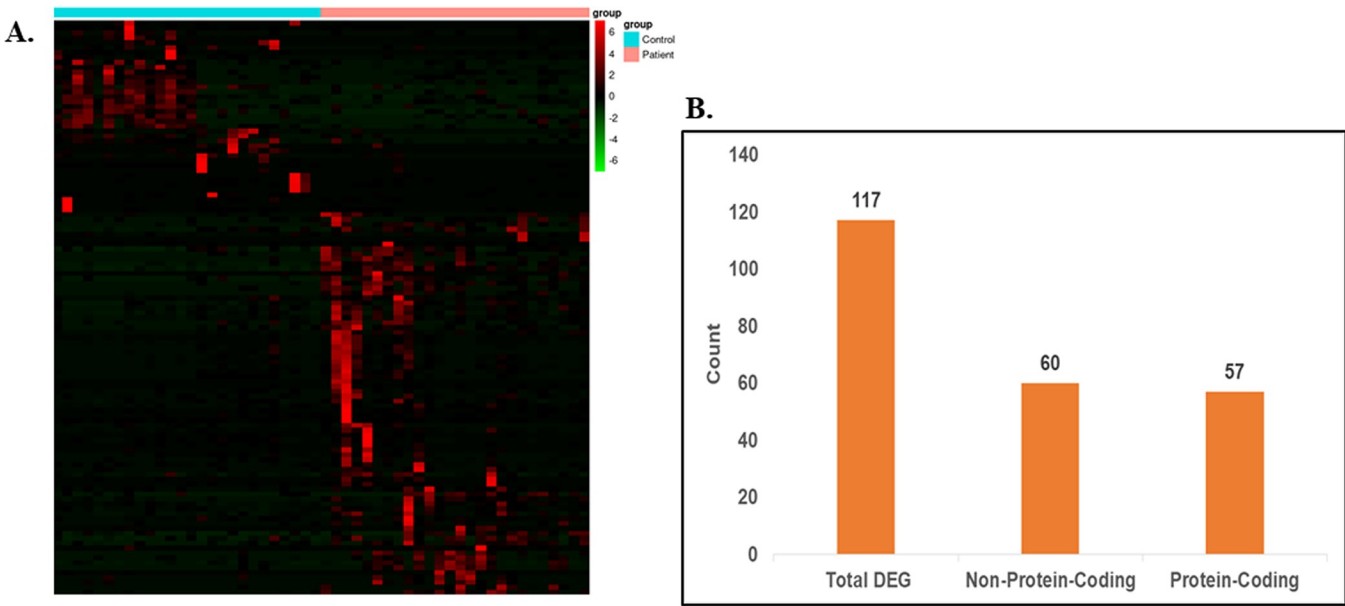

**Fig 1. Differential gene expression analysis of cerebellum tissue samples.** (A) Heatmap of the differentially expressed genes (Pval < 0.01 & FDR < 0.01) in Cerebellum region of the C9ALS patients. (B) Bar plot representing DEGs count.

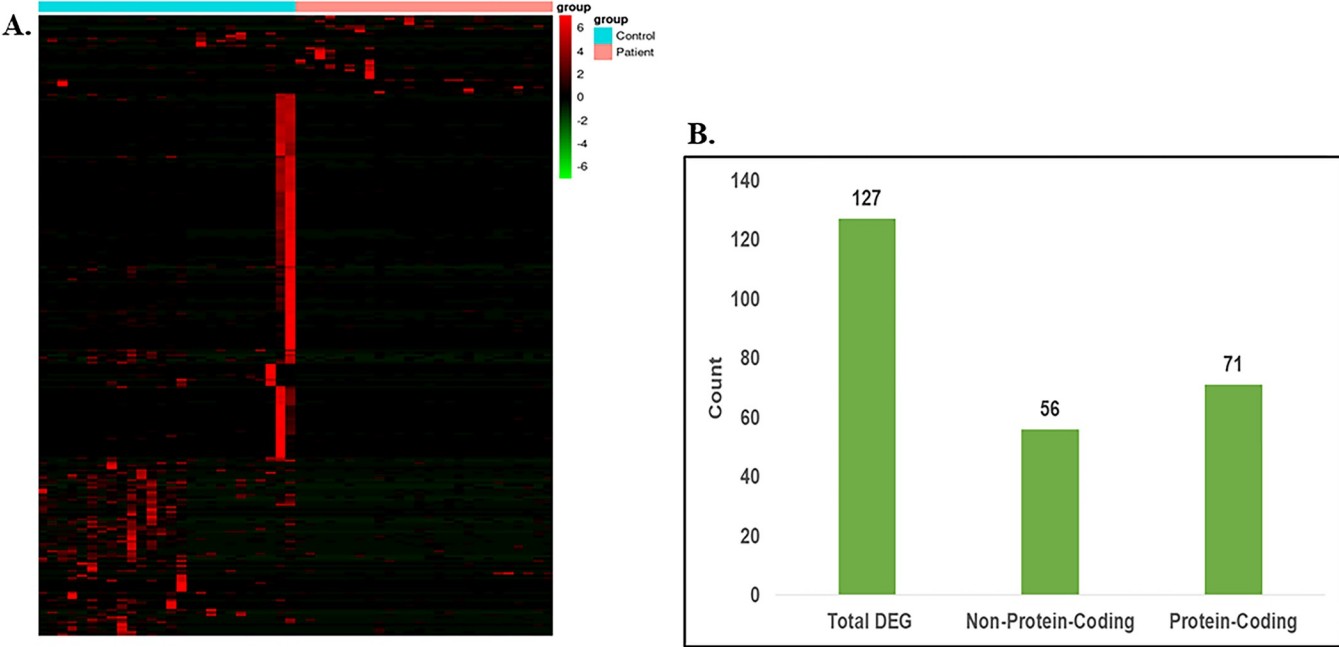

**Fig 2. Differential gene expression analysis of Frontal-cortex tissue samples.** (A) Heatmap of the differentially expressed genes (Pval < 0.01 & FDR < 0.01) in Frontal cortex region of the C9ALS patients. (B) Bar plot representing DEGs count.

DEGs identified in cerebellum tissue samples were mainly enriched in biological process such as peptide secretion, positive regulation of peptide secretion, protein secretion, regulation of cell morphogenesis involved in differentiation, peptide transport, and protein processing along with other processes. In cellular components, the DEGs were mainly enriched in cell surface, platelet alpha granule, external side of plasma membrane, side of membrane, fibrinogen complex, blood microparticle, integral component of membrane and intrinsic component of membrane. Whereas, the molecular functions were enriched in transcription cis regulatory region binding, RNA polymerase II transcription regulatory region sequence specific DNA-binding, transcription regulatory region nucleic acid binding, double stranded DNA binding, and DNA-binding transcription factor activity [Fig 3A] [S3 Table in S1 Data].

Similarly, the GSEA analysis of DEGs identified from frontal cortex samples were mainly enriched in biological processes such as neuron migration, cellular macromolecule metabolic process, neurogenesis, cell migration, neuron generation, locomotion, cell motility and movement of cell or subcellular components. In cellular components, the DEGs were significantly enriched in chromosome, cell periphery, plasma membrane, intrinsic component of plasma membrane, mitochondrion and chromatin locations. Whereas, the molecular functions were mainly enriched in DNA binding, DNA-binding transcription factor activity, sequence specific DNA-binding, transcription regulator activity, and transcription cis-regulatory region binding along with other processes [Fig 4A] [S4 Table in S1 Data].

Further, we also did the pathways enrichment analysis of the DEGs identified from both the tissue samples. The DEGs identified from cerebellum tissue samples were mainly enriched in protein processing in endoplasmic reticulum, platelet activation, legionellosis, longevity regulating pathway, antigen processing and presentation, complement and coagulation cascades, MAPK signalling pathway, estrogen signalling pathway and spliceosome pathway [Fig 3B] [S3 Table in S1 Data]. Whereas the DEGs identified from frontal cortex tissue samples were significantly enriched in neuroactive ligand receptor interaction pathway, glycine, serine and

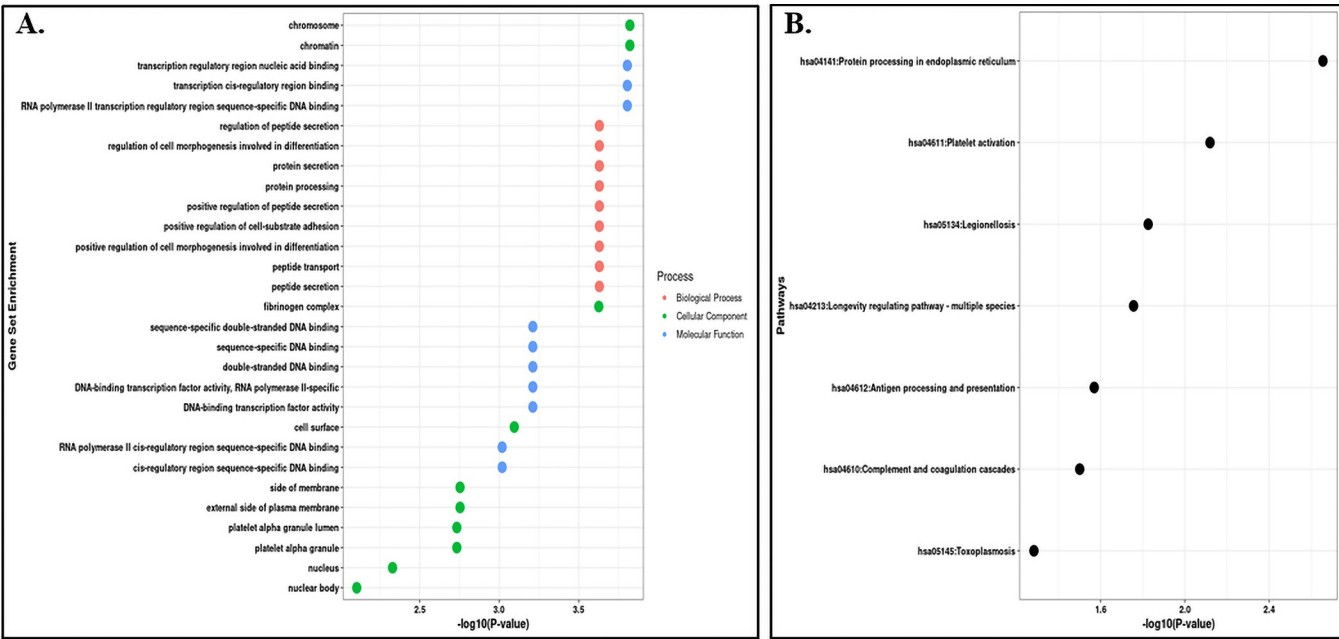

**Fig 3. Gene Ontology and pathway enrichment analysis.** (A) Gene-Set Enrichment Analysis of the dysregulated protein coding genes in the Cerebellum region. (B) Pathways Enrichment Analysis of the dysregulated protein coding genes using KEGG Pathways.

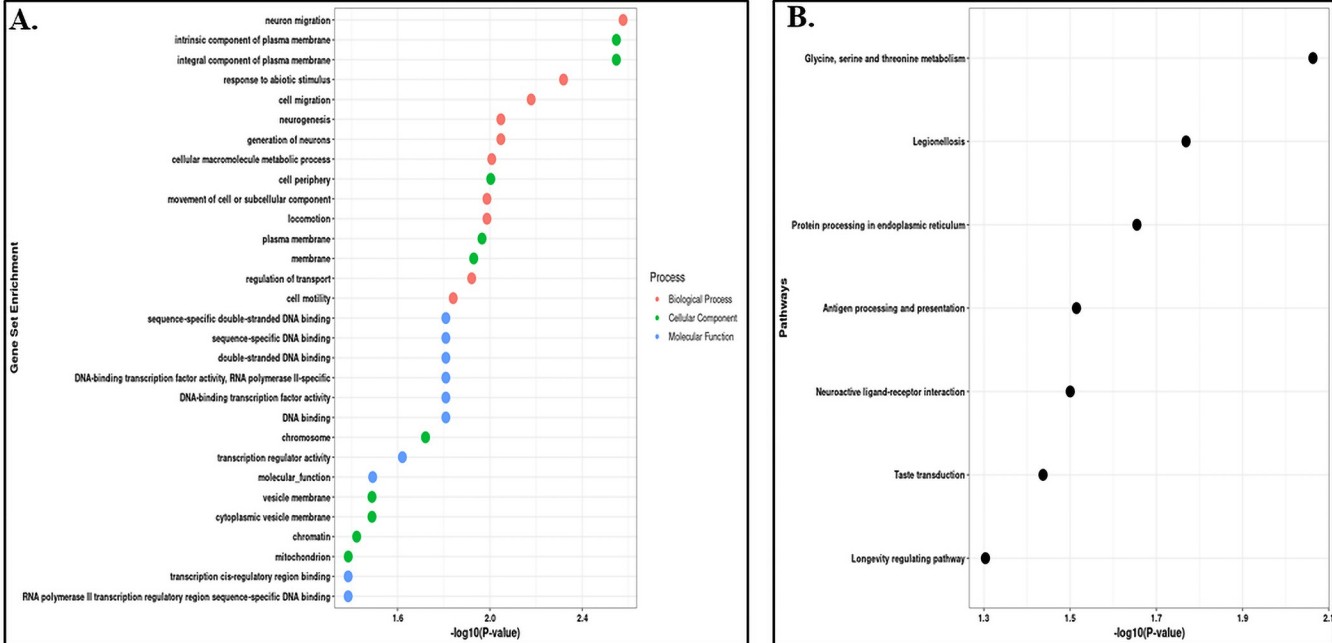

**Fig 4. Gene Ontology and pathway enrichment analysis.** (A) Gene-Set Enrichment Analysis of the dysregulated protein coding genes in the Frontal Cortex region. (B) Pathways Enrichment Analysis of the dysregulated protein coding genes using KEGG Pathways.

threonine metabolism, antigen processing and presentation, and protein processing in endoplasmic reticulum pathways [Fig 4B] [S4 Table in S1 Data].

## Gene-gene and disease-gene interaction network specific to brain

After identifying the genes showing differential expression in cerebellum and frontal cortex tissue samples, genes commonly getting differentially expressed in both the tissues were identified using InteractiVenn, a web-based tool to analyse the gene sets through Venn diagram. A total of three genes, two upregulated (namely HSPA6, and HSPA1A) and one downregulated (namely OPN1SW) were identified in both the tissue samples [Fig 5]. Also, according to the Enrichr results (S5 Table in S1 Data), HSPA6 is differentially expressed in fALS (Familial) whole lumbar spinal cord (GSE52946). HSPA1A is differentially expressed in ALS IPSCs-derived neurons (GSE52202), and in C9orf72 spinal cord knockdown (GSE77681).

List of 2709 genes showing elevated expression in brain was retrieved from the cell atlas database. STRING plug-in of the cytoscape tool was used to prepare the interaction network of genes showing interactions with the three commonly identified genes. A total of 26 genes were making the 107 interactions with the upregulated HSPA6 and HSPA1A genes and among themselves [Fig 6A], whereas only one gene was showing the interaction with the downregulated OPN1SW gene [Fig 6B]. The genes interacting with HSPA6 and HSPA1A gene were showing very high-density interaction with each other, indicating that any changes in the expression of HSPA6 and HSPA1A gene can influence the expression of other genes in this highly connected network.

Apart from gene-gene interaction network, we also prepared the brain specific disease-gene interaction network of the commonly identified DEGs and genes interacting with these DEGs to understand their role in the brain related disorders. A total of 50 brain related disorders such as Glioblastoma, Huntington disease, Dementia, Alzheimer disease late onset, Autism and Mental disorders were found to be associated with these 28 genes, i.e, 2 upregulated genes and 26 other genes being elevated in the brain [Fig 7A] [S5 Table in S1 Data]. Whereas a total of 8 disorders were found to be associated with the downregulated OPN1SW along with one gene GNG13 showing interaction with it [Fig 7B] [S5 Table in S1 Data].

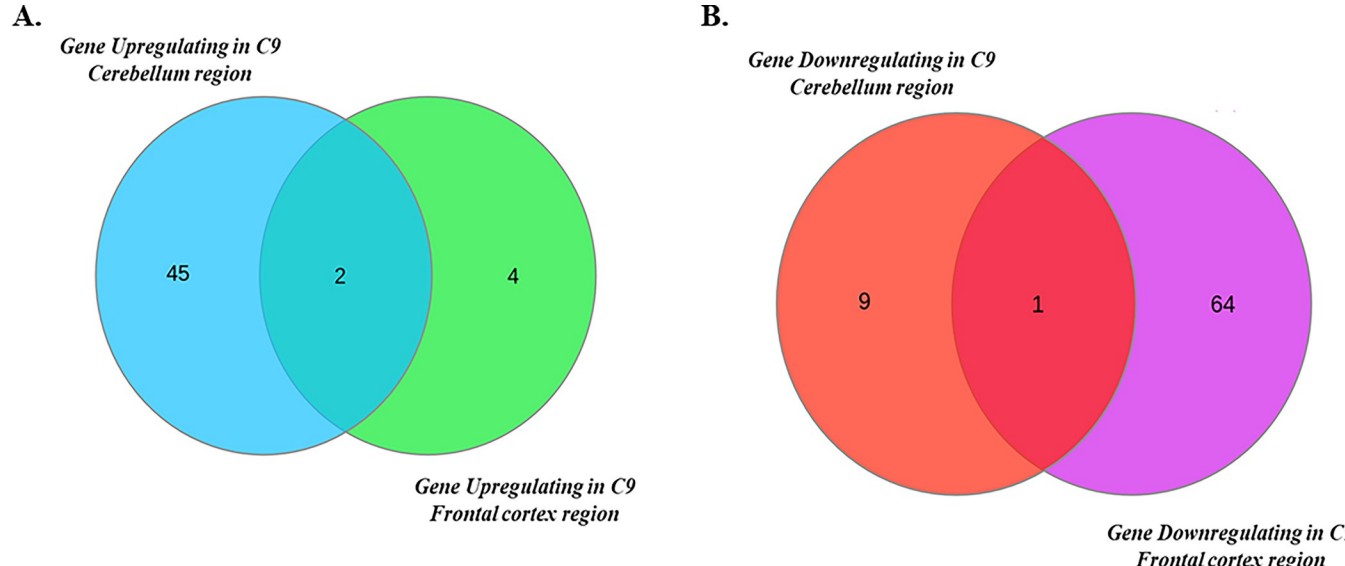

**A.**
Gene Upregulating in C9 Cerebellum region

45 | 2 | 4

Gene Upregulating in C9 Frontal cortex region

**B.**
Gene Downregulating in C9 Cerebellum region

9 | 1 | 64

Gene Downregulating in C9 Frontal cortex region

**Fig 5.** (A) Venn based analysis to identify common upregulated genes in cerebellum and frontal cortex region. (B) Venn based analysis to identify common downregulated genes in cerebellum and frontal cortex region.

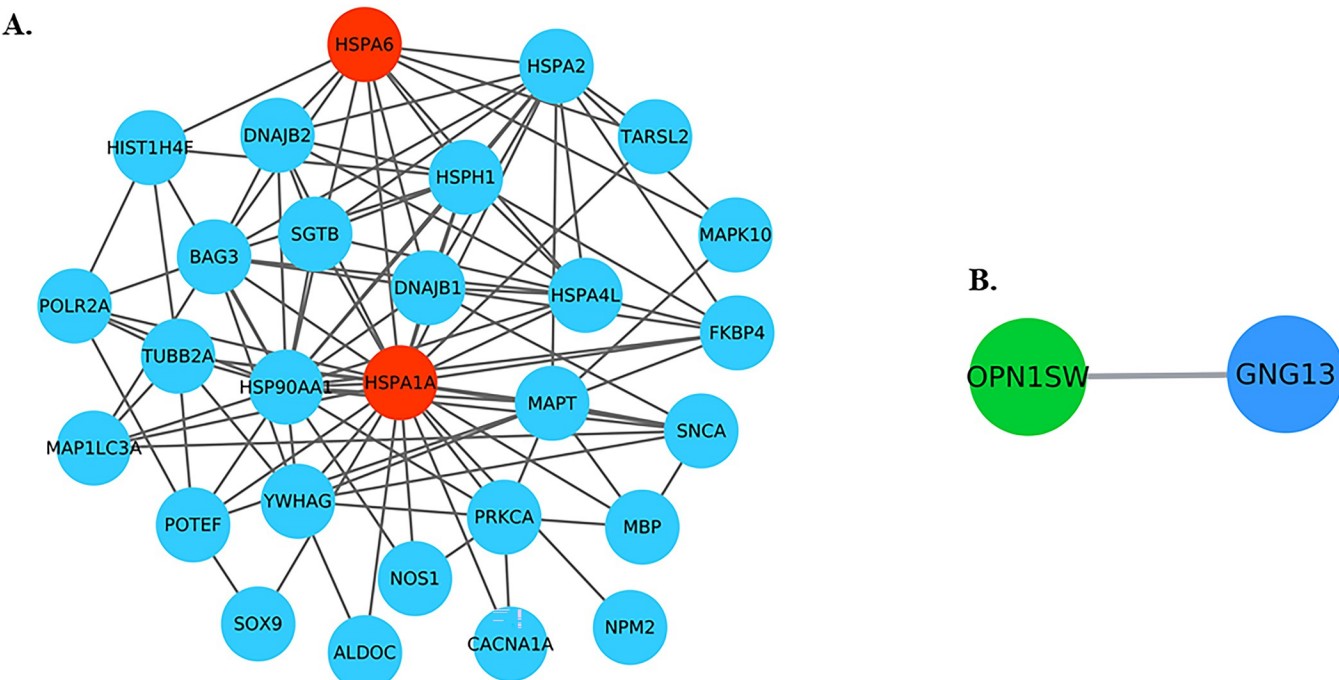

**Fig 6.** (A) Interaction network of common upregulated genes (in red) with the genes elevated in the brain region (in blue). (B) Downregulated OPN1SW gene interacts with only one gene GNG13.

## MicroRNAs as potential regulator of the disease causing differentially expressed genes

MicroRNAs are the small non-coding RNAs having the ability to influence the expression of genes by interacting with their target mRNAs. Several studies have reported miRNAs as potential therapeutic agents. For identifying the miRNAs interacting with our concerned DEGs, several miRNA-gene interaction databases were screened. A total of 118 miRNAs showing

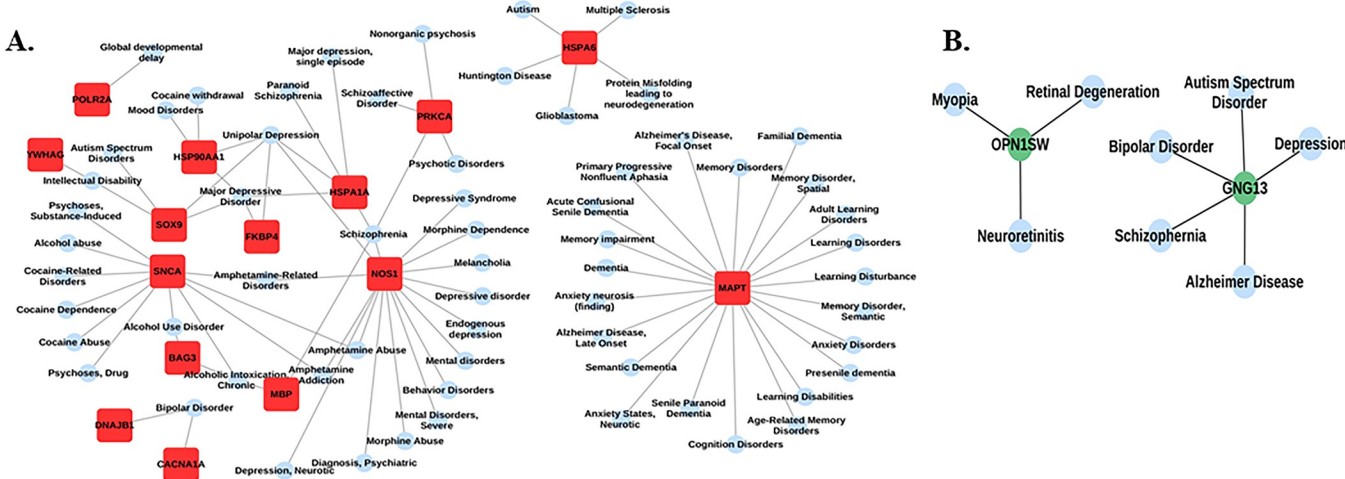

**Fig 7.** (A) Disease-gene interaction network of common upregulated genes along with the interacting genes enriched in brain region. (B) Disease-gene interaction network of common downregulated genes along with the interacting genes enriched in brain region.

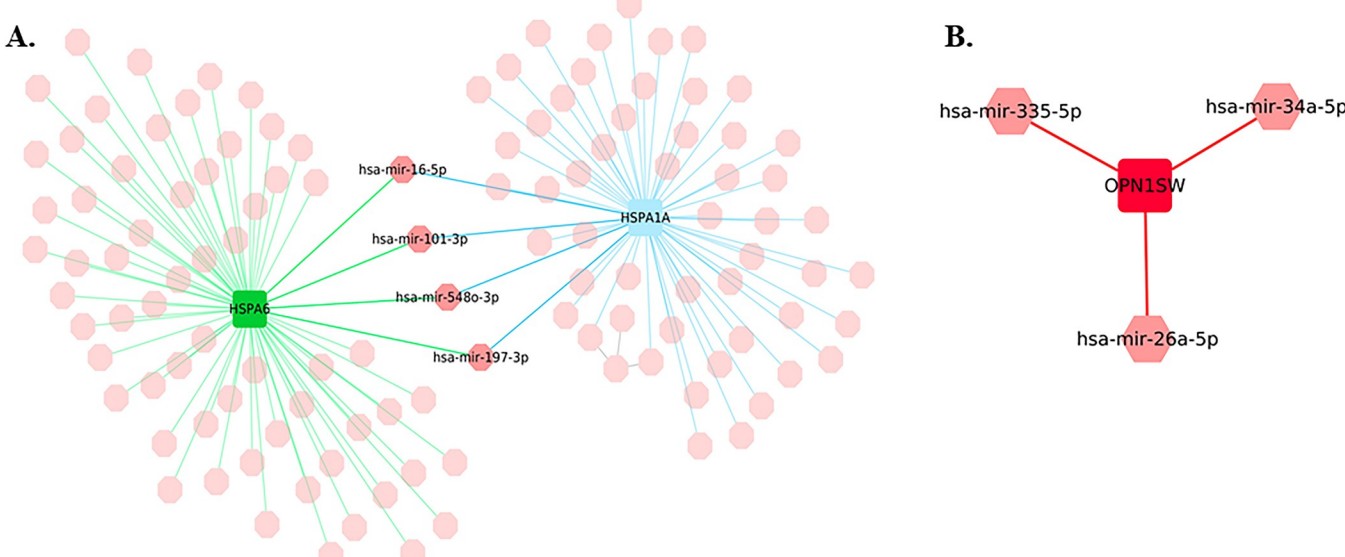

**Fig 8.** (A) miRNA-gene interaction network of common upregulated genes. (B) miRNA-gene interaction network of common downregulated gene (miRNAs are highlighted in pink color nodes).

interaction with HSPA6 and HSPA1A were identified [Fig 8A] [S6 Table in S1 Data]. Only 3 miRNAs showing interaction with OPN1SW were identified [Fig 8B]. Out of 118 identified miRNAs we further try to identify miRNAs interacting with both differentially expressed genes. Further a total of 4 miRNAs namely hsa-miR-16-5p, hsa-miR-101-3p, hsa-miR-5480-3p, and hsa-miR-197-3p were identified being interacted with both the genes. The enrichment analysis of the identified miRNAs reveals their roles in axon cargo development, positive regulation of epidermal growth factor activated receptor activity, negative regulation of translational initiation, neuromuscular process controlling balance, nuclear export, receptor signalling protein serine threonine kinase activity, lipid homeostasis, negative regulation of neuron projection development, motor activity, and locomotory behaviour [Fig 9A] [S7 Table in S1 Data]. Pathway's enrichment analysis revealed their role in Notch signalling pathway, Oxytocin receptor mediated signalling pathway, Aminoacyl tRNA biosynthesis, MAPK cascade, cysteine and methionine metabolism, Amyotrophic lateral sclerosis pathway, Parkinson's disease pathway, VEGF signalling pathway, TGF beta signalling pathway, spliceosome, Alzheimer's disease, FGF signalling pathway and apoptosis signalling pathways [Fig 9B] [S7 Table in S1 Data].

## Conclusion

In this manuscript, we have identified that the genes involved in neuron development, protein localization and transcription are mostly enriched in cerebellum of C9-ALS patients, while the UPR-related genes are enriched in the frontal cortex. Of note, UPR pathway genes were mostly dysregulated both in the C9-ALS cerebellum and frontal cortex. Overall, the data presented here show that defects in normal RNA processing and the UPR pathway are the pathological hallmarks of C9-ALS. Interestingly, the cerebellum region showed more significant transcriptomic changes as compared to the frontal cortex region suggesting its active participation in the disease process. This nuanced understanding may offer valuable insights for the development of targeted therapeutic strategies aimed at mitigating disease progression in C9-ALS.

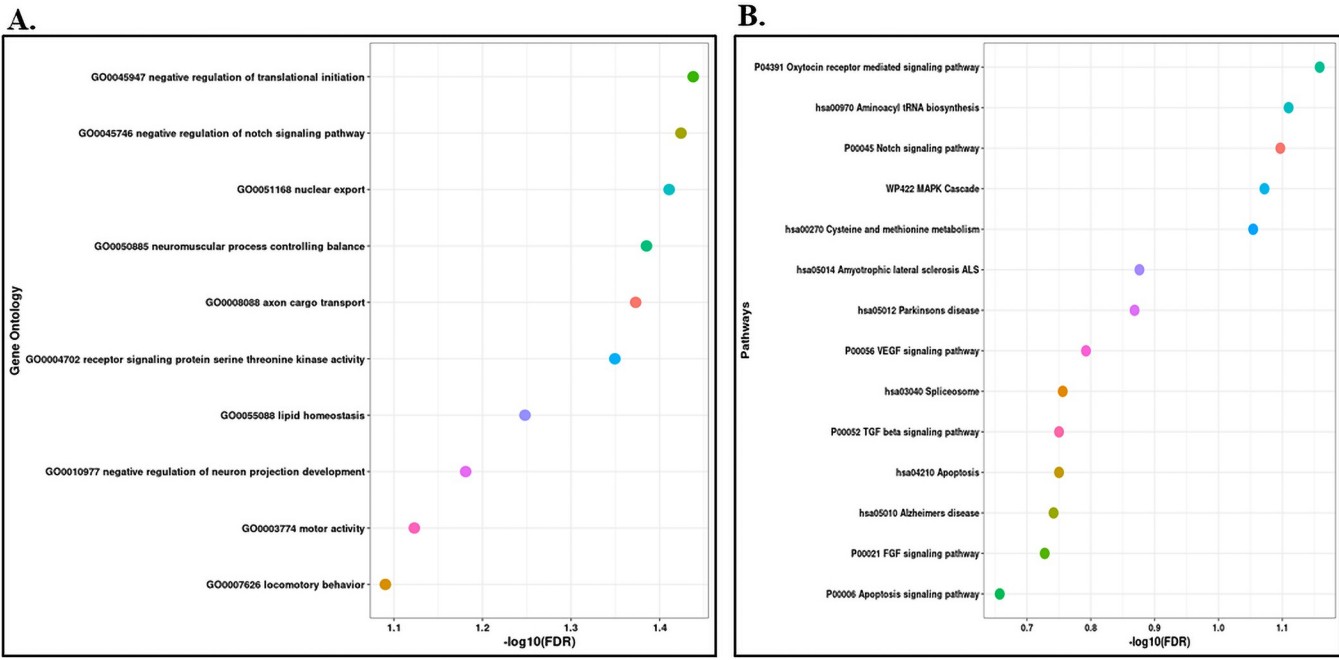

**Fig 9.** (A) Gene-Ontology analysis of miRNAs interacting with the upregulated genes. (B) Pathways enrichment analysis of the interacting miRNAs.

## Supporting information

**S1 Data. List of the differentially expressed genes (DEGs) and their functional enrichment in cerebellum and frontal cortex region of C9-ALS.** List of miRNAs interacting with key DEGs and their gene enrichment.
(XLSX)

## Acknowledgments

The authors sincerely thank Amity University, Noida for providing facilities. The authors extend their appreciation to the Department of Science and Technology, Government of India.

## Author Contributions

**Conceptualization:** Kartikay Prasad.

**Data curation:** Kartikay Prasad, Saurabh Raghuvanshi.

**Methodology:** Kartikay Prasad.

**Software:** Kartikay Prasad.

**Supervision:** Saurabh Raghuvanshi, Vijay Kumar.

**Visualization:** Kartikay Prasad.

**Writing – original draft:** Kartikay Prasad, Vijay Kumar.

**Writing – review & editing:** Md Imtaiyaz Hassan, Vijay Kumar.

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
