## [Decision Letter · Decision Letter 0]

12 Feb 2024

PONE-D-23-33311Understanding the Relationship between Cerebellum and the Frontal-Cortex region of c9orf72-Related Amyotrophic Lateral Sclerosis: A Comparative Analysis of Genetic FeaturesPLOS ONE

Dear Dr. Kumar,

Thank you for submitting your manuscript to PLOS ONE. After careful consideration, we feel that it has merit but does not fully meet PLOS ONE’s publication criteria as it currently stands. Therefore, we invite you to submit a revised version of the manuscript that addresses the points raised during the review process.

We look forward to receiving your revised manuscript.

Kind regards,

Massimo Filippi

Academic Editor

PLOS ONE

https://academic.oup.com/bib/article-abstract/23/6/bbac442/6780269?redirectedFrom=fulltext&login=false

https://pubmed.ncbi.nlm.nih.gov/32905541/

In your revision ensure you cite all your sources (including your own works), and quote or rephrase any duplicated text outside the methods section. Further consideration is dependent on these concerns being addressed.

4. We suggest you thoroughly copyedit your manuscript for language usage, spelling, and grammar. If you do not know anyone who can help you do this, you may wish to consider employing a professional scientific editing service. 

A clean copy of the edited manuscript (uploaded as the new *manuscript* file)”.

6. Thank you for stating the following financial disclosure: 

 [INDIA - IN].  

7. Thank you for stating the following in the Acknowledgments Section of your manuscript: 

[K.P. sincerely thanks the Indian Council of Medical Research (ICMR) New Delhi, India for providing the Senior Research Fellowship grant (BMI/11(63)/2020). VK sincerely thanks the Indian Council of Medical Research (ICMR) New Delhi, India for the financial support (DHR/Neuro/2020-NCD-I). The authors sincerely thank Amity University, Noida for providing facilities. The authors extend their appreciation to the Department of Science and Technology, Government of India.]

 [INDIA - IN]. 

Reviewers' comments:

Reviewer's Responses to Questions

**Comments to the Author**

1. Is the manuscript technically sound, and do the data support the conclusions?

Reviewer #1: Yes

Reviewer #2: Yes

2. Has the statistical analysis been performed appropriately and rigorously? 

Reviewer #1: Yes

Reviewer #2: Yes

3. Have the authors made all data underlying the findings in their manuscript fully available?

Reviewer #1: Yes

Reviewer #2: Yes

4. Is the manuscript presented in an intelligible fashion and written in standard English?

Reviewer #1: No

Reviewer #2: Yes

5. Review Comments to the Author

Reviewer #1: This manuscript reports the results of a study assessing transcriptomic features of biological samples obtained from biobank of brain cortical tissue of 25 C9-ALS patients and healthy controls. Gene expression analysis and biological characterization of identified genes of interest were performed, as well as screening of possible miRNAs to interact with such genes.

Although the study is of interest, and methodology seems overall adequate, I have important concerns regarding the clarity of the manuscript in its present form, which looks partially incomplete, disorganized and slightly out of focus as regards the clinical relevance of the authors' findings.

-Introduction reports several details about FTD, but here the authors included only patients with ALS. I would limit mentioning FTD.

- The authors state that "frozen human post-mortem tissue was obtained from motor cortex (medial and lateral)", but the manuscript only reports data from frontal and cerebellar tissue.

-The authors state that data were obtained from "25 C9-ALS patients and control patients", but it is unclear how many were C9 and how many were controls. Moreover, what do the authors mean as "control patients"? Were they healthy individuals or patients with different diseases? This should be clearly specified.

-In some paragraphs of the methods section the authors use the future tense (e-g., "the count data will then further be used"). This should be corrected.

-Some paragraphs of the results section actually report details to be reported in the methods. See for example "DESeq2 tool was then used to ...".

- The quality of English should be reviewed carefully. See for example "the current study compared the transcriptome changes in the cerebellum and frontal cortex of C9-ALS cases, with between (?), indicating etc.")

- Why did the authors only select frontal and cerebellar regions? This should be better clarified.

Reviewer #2: The aim of this study was identifying and and spatially determining differential gene expression signature differences between cerebellum and frontal cortex in C9ALS, to study the network properties of these differentially expressed genes, and to identify miRNAs targeting the common differentially expressed genes in both the tissues. The study used data generated by several research groups.

The authors observed that the genes involved in neuron development, protein localization and transcription are mostly enriched in cerebellum of C9ALS patients, while the UPR-related genes are enriched in the frontal cortex. Finally, UPR pathway genes were mostly dysregulated both in the C9ALS cerebellum and frontal cortex.

The study is well performed and highlight a possible role of cerebellum in the neurodegenerative process of C9ALS.

The fina sentence of the paper (Although the cerebellum role in coordination and motor control is well established, its involvement in ALS has been largely unnoticed) is not correct. Just limiting on cases with C9ORF72 expansion, the role of cerebellum has been identified in clinical (PMID: 35189395), neuropathological (PMID 37816685), neuroimaging (PMID: 34168085; PMID: 34544819), machine learning (PMID: 36694130) studies as well as in preclinical models (PMID: 35843530). In addition, the presence of cerebellar functional involvement has been found to be a prognostic predictor in ALS (PMID: 36308536). I suggest commenting these important papers at the light of the findings of the present study.

6. PLOS authors have the option to publish the peer review history of their article (what does this mean?). If published, this will include your full peer review and any attached files.

Reviewer #1: No

Reviewer #2: No

---

## [Author Response · Author response to Decision Letter 0]

24 Feb 2024

Response to reviewers' comments

Manuscript title: Understanding the Relationship between Cerebellum and the Frontal-Cortex region of C9orf72-Related Amyotrophic Lateral Sclerosis: A Comparative Analysis of Genetic Features

Ms. Ref. No.: PONE-D-23-33311

Referee(s)' Comments to Author:

Reviewer 1:

This manuscript reports the results of a study assessing transcriptomic features of biological samples obtained from biobank of brain cortical tissue of 25 C9-ALS patients and healthy controls. Gene expression analysis and biological characterization of identified genes of interest were performed, as well as screening of possible miRNAs to interact with such genes. Although the study is of interest, and methodology seems overall adequate, I have important concerns regarding the clarity of the manuscript in its present form, which looks partially incomplete, disorganized, and slightly out of focus as regards the clinical relevance of the authors' findings.

 Comment 1: Introduction reports several details about FTD, but here the authors included only patients with ALS. I would limit mentioning FTD.

Response: We thank the reviewer for reading our manuscript critically and providing the valuable comments. We agreed to the reviewer comment and edited the manuscript accordingly.

Comment 2: The authors state that "frozen human post-mortem tissue was obtained from motor cortex (medial and lateral)", but the manuscript only reports data from frontal and cerebellar tissue.

Response: Thank you for pointing out this mistake. The manuscript has been corrected.

Comment 3: The authors state that data were obtained from "25 C9-ALS patients and control patients", but it is unclear how many were C9 and how many were controls. Moreover, what do the authors mean as "control patients"? Were they healthy individuals or patients with different diseases? This should be clearly specified.

Response: Manuscript has been edited as per suggestion to give clearer idea about the samples. The control patient term represented healthy individuals, the same has been edited in the manuscript. 

Comment 4: In some paragraphs of the methods section the authors use the future tense (e-g., "the count data will then further be used"). This should be corrected.

Response: We have modified the sentences in the revised manuscript. 

Comment 5: Some paragraphs of the results section actually report details to be reported in the methods. See for example "DESeq2 tool was then used to ...".

Response: The repetition has been removed as per suggestion. 

Comment 6: The quality of English should be reviewed carefully. See for example "the current study compared the transcriptome changes in the cerebellum and frontal cortex of C9-ALS cases, with between (?), indicating etc.")

Response: A large junk of manuscript has been modified accordingly and we highly appreciate their contributions in shaping the manuscript as it is now. 

Comment 7: Why did the authors only select frontal and cerebellar regions? This should be better clarified.

Response: We have incorporated a brief description of the importance of frontal and cerebellar regions in ALS in the revised manuscript.

Reviewer 2: 

The aim of this study was identifying and spatially determining differential gene expression signature differences between cerebellum and frontal cortex in C9ALS, to study the network properties of these differentially expressed genes, and to identify miRNAs targeting the common differentially expressed genes in both the tissues. The study used data generated by several research groups. The authors observed that the genes involved in neuron development, protein localization and transcription are mostly enriched in cerebellum of C9ALS patients, while the UPR-related genes are enriched in the frontal cortex. Finally, UPR pathway genes were mostly dysregulated both in the C9ALS cerebellum and frontal cortex.

The study is well performed and highlight a possible role of cerebellum in the neurodegenerative process of C9-ALS.

The final sentence of the paper (Although the cerebellum role in coordination and motor control is well established, its involvement in ALS has been largely unnoticed) is not correct. Just limiting on cases with C9ORF72 expansion, the role of cerebellum has been identified in clinical (PMID: 35189395), neuropathological (PMID 37816685), neuroimaging (PMID: 34168085; PMID: 34544819), machine learning (PMID: 36694130) studies as well as in preclinical models (PMID: 35843530). In addition, the presence of cerebellar functional involvement has been found to be a prognostic predictor in ALS (PMID: 36308536). I suggest commenting these important papers at the light of the findings of the present study.

Response: We thank reviewer for reading our manuscript closely and providing the valuable comment. As per the reviewer suggestions we have discussed the mentioned and cited the articles in our manuscript and also corrected the statement which reviewer suggested.

---

## [Decision Letter · Decision Letter 1]

14 Mar 2024

Understanding the Relationship between Cerebellum and the Frontal-Cortex region of C9orf72-Related Amyotrophic Lateral Sclerosis: A Comparative Analysis of Genetic Features

PONE-D-23-33311R1

Dear Dr. Kumar,

We’re pleased to inform you that your manuscript has been judged scientifically suitable for publication and will be formally accepted for publication once it meets all outstanding technical requirements.

Kind regards,

Massimo Filippi

Academic Editor

PLOS ONE

Additional Editor Comments (optional):

Reviewers' comments:

Reviewer's Responses to Questions

**Comments to the Author**

1. If the authors have adequately addressed your comments raised in a previous round of review and you feel that this manuscript is now acceptable for publication, you may indicate that here to bypass the “Comments to the Author” section, enter your conflict of interest statement in the “Confidential to Editor” section, and submit your "Accept" recommendation.

Reviewer #1: All comments have been addressed

Reviewer #2: All comments have been addressed

2. Is the manuscript technically sound, and do the data support the conclusions?

Reviewer #1: Yes

Reviewer #2: Yes

3. Has the statistical analysis been performed appropriately and rigorously? 

Reviewer #1: Yes

Reviewer #2: Yes

4. Have the authors made all data underlying the findings in their manuscript fully available?

Reviewer #1: Yes

Reviewer #2: Yes

5. Is the manuscript presented in an intelligible fashion and written in standard English?

Reviewer #1: Yes

Reviewer #2: Yes

6. Review Comments to the Author

Reviewer #1: I have no further comments. The authors have addressed my previous concerns regarding the rationale of selecting some cortical regions

Reviewer #2: All my comments were adequately addressed. The paper can be published in the current form. Congratulations to the authors for their work.

7. PLOS authors have the option to publish the peer review history of their article (what does this mean?). If published, this will include your full peer review and any attached files.

Reviewer #1: No

Reviewer #2: No

---

## [Editor Report · Acceptance letter]

22 Mar 2024

PONE-D-23-33311R1 

PLOS ONE

Dear Dr. Kumar, 

I'm pleased to inform you that your manuscript has been deemed suitable for publication in PLOS ONE. Congratulations! Your manuscript is now being handed over to our production team.

Kind regards, 

on behalf of

Prof. Massimo Filippi 

Academic Editor

PLOS ONE